# The Role of Ultrasound Examination in the Assessment of Suitability of Calcified Arteries for Vascular Access Creation—Mini Review

**DOI:** 10.3390/diagnostics13162660

**Published:** 2023-08-12

**Authors:** Jakob Gubensek

**Affiliations:** 1Center for Acute and Complicated Dialysis and Vascular Access, Department of Nephrology, University Medical Center Ljubljana, 1000 Ljubljana, Slovenia; jakob.gubensek@kclj.si; Tel.: +386-1-522-3112; Fax: +386-1-522-2292; 2Faculty of Medicine, University of Ljubljana, 1000 Ljubljana, Slovenia

**Keywords:** arterio-venous fistula, ultrasound, mapping, arterial calcifications, mediocalcinosis, Doppler, outcomes

## Abstract

Arterial calcifications are present in 20–40% of patients with end-stage kidney disease and are more frequent among the elderly and diabetics. They reduce the possibility of arterio-venous fistula (AVF) formation and maturation and increase the likelihood of complications, especially distal ischemia. This review focuses on methods for detecting arterial calcifications and assessing the suitability of calcified arteries for providing inflow before the construction of an AVF. The importance of a clinical examination is stressed. A grading system is proposed for quantifying the severity of calcifications in the arteries of the arm with B-mode and Doppler ultrasound exams. Functional tests to assess the suitability of the artery to provide adequate inflow to the AVF are discussed, including Doppler indices (peak systolic velocity and resistive index during reactive hyperemia). Possible predictors of the development of distal ischemia are discussed (finger pressure, digital brachial index, acceleration and acceleration time), as well as the outcomes of AVFs placed on calcified arteries. It is concluded that a noninvasive ultrasound examination is probably the best tool for a morphologic and functional assessment of the arteries. An arterial assessment is of utmost importance if we are to create distal radiocephalic AVFs in our elderly patients whenever possible without burdening them with futile surgical attempts.

## 1. Introduction

Arterial calcifications are common in patients with advanced stages of chronic kidney disease (CKD). Among them, they are most common in patients with end-stage kidney disease (ESKD) on dialysis. In addition to the stage of CKD, other risk factors for arterial calcification include advanced age, diabetes, smoking, deranged CKD-related mineral and bone disease, as well as hypertension and hyperlipidemia [1,2,3]. The presence of arterial calcifications worsens the prognosis of patients with CKD and increases the likelihood of cardiovascular events [4,5]. Furthermore, arterial calcifications also reduce the possibility of arterio-venous fistula (AVF) formation and maturation and increase the likelihood of complications, especially distal ischemia [6,7,8,9]. This is a significant disadvantage, as an AVF is the optimal vascular access for patients on maintenance hemodialysis and is thus their lifeline. 

A precise evaluation of the suitability of a calcified artery for possible creation of an AVF remains a challenging task. This task will become increasingly more important in the future as the ESKD population is getting older and calcifications become even more prevalent. Furthermore, the decision to perform a forearm AVF instead of an elbow AVF whenever possible, even in the presence of calcifications, is not only an academic or policy issue because forearm AVFs have much lower frequency of ischemic complications. Therefore, an arterial assessment is of utmost importance if we are to create distal radiocephalic AVFs in our elderly patients whenever possible without burdening them with futile surgical attempts. 

This review is intended for nephrologists, working in the field of vascular access and interventional nephrology, and focuses on methods to detect arterial calcifications and to assess the suitability of calcified arteries for providing inflow before the construction of an AVF. It gives a critical appraisal of existing methods and some comparison between the fields of vascular access and peripheral arterial disease. An ultrasound examination is the main tool we have, and a grading system is proposed for quantifying the severity of calcifications in the arteries of the arm with B-mode and Doppler ultrasound exams. Some of Doppler parameters described in the literature, which can be used for arterial assessment, are discussed, and the best parameters that can be used in clinical practice are emphasized. Possible predictors of the development of distal ischemia are also discussed, as well as the overall outcomes of AVFs placed on calcified arteries. 

## 2. Prevalence of Calcifications in the Arteries of the Arm

In a large biopsy study of radial arteries harvested during AVF construction in 180 ESKD patients, calcifications in the media were reported in 21% of patients [2]. In 12% of patients, calcifications were graded as mild or moderate, while in the remaining 9% they were severe [2]. A smaller biopsy study of 30 ESKD patients reported an even higher prevalence of calcifications in the radial artery, which were mild or moderate in 20% and severe in 17% [3]. Very precise detection with histomorphometry revealed an even higher prevalence of 40% microcalcifications in the media (defined as >1% of the media being calcified), but this did not affect the outcome of AVFs [10]. 

Although some large contemporary cohorts on preoperative vascular mapping do not report the presence of arterial calcifications [11,12], the ones that do report calcifications report similar prevalence rates. Srivastava reported a cohort of 173 Indian patients, among whom arterial calcifications were observed in 15 diabetic patients, resulting in a 9% prevalence of calcifications in the radial artery [13]. Our group reported a much higher 20% prevalence of moderate or severe calcifications in 129 elderly (>65 years) patients referred for mapping, of whom 37% were diabetic [14]. Severe calcifications can preclude the formation of an AVF. Cho published the results of a vascular mapping of a large cohort of 299 patients with a mean age of 62 years, of whom 48% were diabetic, and reported that a radiocephalic AVF was not possible in 5% solely due to arterial wall calcifications or stenosis and that some patients had combined reasons for the impossibility of distal AVF creation [15]. 

Arterial calcifications therefore represent a significant clinical problem among elderly and diabetic patients and influence the decision of where to place an AVF. With the improved survival of CKD patients, we are faced with the increasing age and comorbidity of patients reaching ESKD and requiring vascular access for hemodialysis. Therefore, the size of the problem will increase even further in the future.

## 3. Summary of Current Recommendations 

The guidelines on vascular access in dialysis patients do not provide precise criteria for assessing the suitability of an artery prior to AVF creation beyond a cut-off value for the diameter. The 2019 KDOQI vascular access guideline recommends clinical examinations in all patients and an ultrasound mapping in patients at high risk for AVF failure, which includes patients with a clinically evident atherosclerosis and the elderly [16]. They find it reasonable to evaluate multiple characteristics of vessel quality and resistance index (RI) during reactive hyperemia is mentioned as one of them in the rationale section [16], based on the study by Malovrh [17]. On the other hand, guidelines from European Society for Vascular Surgery recommend the preoperative ultrasonography of bilateral upper extremity arteries and veins in all patients [6], since randomized trials showed that an ultrasound examination enhances the success of the creation and outcomes of autogenous AVFs [18,19]; further studies supporting this were published after the publication of the guidelines [20]. It is mentioned that vessel calcification may limit vascular access options, particularly in diabetic patients, but not how the decision should be made [6]. They state that an arterio-venous anastomosis can still be performed to arteries with a mild “eggshell” calcification, while a severe calcification makes performing the anastomosis difficult, and the associated vessel rigidity may compromise maturation [6]. The new European guidelines completely avoid the presurgical assessment issues [21]. To summarize, the guidelines seem to suggest that severe calcifications significantly reduce the success of AVFs and imply that this may be at least a relative contraindication for the selection of a specific site for AVF formation. Furthermore, RI during reactive hyperemia is mentioned as a possible criterion with a cut-off value of >0.70, based on the study by Malovrh [17].

## 4. How to Assess Calcified Arteries Prior to Vascular Access Creation

It was shown that a preoperative ultrasound examination of the arteries often significantly changes the surgical plan, particularly when a radiocephalic AVF is planned [22]. What we look for in the arterial assessment is not only the presence of calcifications by themselves, but also a functional assessment of the suitability of the artery to provide inflow to the AVF. Therefore, the question that must be answered by the clinical and ultrasound examination is: if an adequate vein is available, when should a calcified radial artery be declared too calcified and therefore an AVF creation attempt futile?

### 4.1. Methods for Detection and Quantification of Calcifications

#### 4.1.1. Clinical Examination

When palpating the pulse of an artery, especially in nonobese patients in whom the amount of tissue between the skin and the artery is small, severe calcifications of the arterial wall can be felt. The wall of a calcified artery is felt as a hard, irregular, lumpy surface, and in the most severe cases, no pulse is felt as the artery can no longer be compressed. 

The significance of a palpable arterial wall was observed in a study from India [7]. The wall of the radial artery was palpable in 26 of 66 patients, while the artery had a linear calcification on the radiograph of the forearm in only 24 patients, 9 of whom had linear calcifications [7]. From this it would seem the authors reported a “palpable” artery even with relatively mild calcifications. Nevertheless, palpable arterial walls proved to be a significant predictor of primary AVF failure in a multivariate model, with an odds ratio (OR) of 15.3 (95% confidence interval (CI) 2.5–62.6, *p* = 0.003) on a background of a relatively poor 3-month primary failure rate of 47% [7].

#### 4.1.2. Plain Radiograph

Plain radiographs, mainly lateral abdominal and pelvic radiographs (performed as a part of cystogram), are recommended as a screening tool for detecting calcifications of large arteries in hemodialysis patients. Similarly, hand radiographs are routinely used in some centers, mainly by vascular surgeons, to screen for arterial calcifications in the radial/brachial arteries prior to AVF creation, particularly in diabetic patients [7,9]. The characteristic “railroad track” pattern of radial artery media wall calcification is seen on the radiograph (see Figure 1).

The calcifications can be graded as spotty/nonlinear or diffuse/linear and their extent can be further quantified by the length of the radial artery that is calcified. Georgiadis proposed a classification of the length of linear calcifications on a radiograph as < or >6 cm [9], which was later also used in other studies [7]. As calcifications are known to have a peripheral distribution, this distinction roughly corresponds to a limited calcified segment of a distal radial artery of the wrist vs. more severe and proximal calcifications. Using this classification, Georgiadis reported nearly halved primary and secondary patency rates compared with absent calcifications or calcifications of <6 cm in length [9]. In a study by Suresh Kumar, a linear radial artery calcification of ≥6 cm in length was also significantly associated with primary AVF failure in a multivariable model with an OR of 7.9 (95% CI 1.5–42.7), while nonlinear (<6 cm in length) calcifications were not [7]. 

The problem with this classification is that it does not quantify the severity of the calcification at the site of planned anastomosis (i.e., predicting the difficulty/impossibility of a clamping/surgical procedure), but rather provides information about the overall severity of calcifications, which is associated with the ability to provide inflow and therefore with maturation success.

#### 4.1.3. B-Mode Ultrasound

Ultrasound mapping of vessels is considered as a standard of care for the preoperative evaluation of arteries (and veins) in the developed world. Using B-mode ultrasounds, calcifications of the arterial wall can readily be seen and also quantified. There is no commonly accepted grading scale for quantifying calcifications in upper extremity arteries. The American Institute of Ultrasound in Medicine guideline for the performance of ultrasound vascular mapping only requires that the presence of calcifications is recorded and reported but does not require the level of calcifications to be graded nor do they provide a grading scale [23]. Nevertheless, there are several grading scales used in the literature. The definitions of moderate or severe calcifications used in the literature are inconsistent and usually skewed toward detecting either very mild or very severe calcifications. For example, the study by Allon [10] seems to focus more on microcalcifications, as even “severe calcifications” in their definition do not make a significant distal shadow. Similarly, a study on femoral arteries found that an ultrasound was more sensitive than conventional radiography for detecting medial calcifications [24]. Here, the highest score was defined as a calcification of “only” >3 cm in length [24]. Some of the comprehensive scoring systems that aim to make a general assessment of the extent of calcifications assess the extent of calcifications by estimating their length in selected arteries (i.e., the length method) and assign higher scores for longer calcified segments [24,25]. The other possibility is to assign points for the presence of calcifications in each of the assessed segments of the arterial tree, regardless of their length or intensity (i.e., the segmentation method) [25]. A comparison of segmentation and length methods found that only the segmentation method scores were independent predictors of peripheral vascular disease and diabetic nephropathy [25]. Furthermore, the presence of calcifications in the radial artery (not graded) was shown to predict the functional maturation of radiocephalic AVFs [13].

However, the intensity of the calcifications might be more important in assessing the suitability of the artery for the creation of vascular access, as severe circumferential calcifications make the artery incompressible for clamping, making the creation of the anastomosis much more difficult and the outcomes much worse. A grading system has been proposed and validated for the assessment of the severity of calcification in the superficial femoral artery [26]. Based on this grading system, clinical experience and other published literature [1,17,27], a semiquantitative calcification grading scale can also be proposed for arteries of the arm (see Table 1 and Figure 2). 

#### 4.1.4. Compressibility of the Artery

Although vascular surgeons have techniques to make anastomoses to completely rigid arteries [28,29], most access surgeons want the feeding artery to be compressible when performing vascular access surgery, both for the ease of the procedure and for improved outcomes of AVF creation. The compressibility of a calcified artery can be assessed prior to surgery. While viewing the artery in a cross-sectional view, the ultrasound probe can be pressed at an appropriate angle so that it compresses the artery onto the bone (radius) [27]. The compressibility should be tested at several sites in the area of the desired anastomosis, as calcifications are focal (see Appendix A). This test mimics the clamping of the artery during surgery and can likely prevent unnecessary surgery if the surgeon is not prepared to deal with a completely rigid artery.

#### 4.1.5. Color Doppler Assessment

Different color and pulsed-wave Doppler parameters can be used to assess the suitability of the artery for AVF creation (see Table 1 and Figure 2E,F). First, the homogeneity of the color Doppler signal can be assessed. Sonographic drop outs, which are visible as gaps in the color Doppler signal, are caused by dense calcifications that also cause distal shadowing [30,31]. The homogeneity or continuity of the color Doppler signal within the arterial lumen can therefore be used to assess the severity of the observed arterial wall calcifications when they are present [27]. Whether this improves the assessment of the severity of calcifications compared with an assessment using B-mode imaging alone remains to be determined.

### 4.2. Functional Assessment of the Artery

For an AVF to mature, the blood flow through the feeding artery must increase by more than 10 times. This is achieved by the dilatation of the artery in response to increased wall shear stress resulting from increased velocities. It can be expected that arterial dilation will be insufficient in the case of a severe and circumferential calcification of the artery, especially if the calcifications involve a long segment of the artery. Therefore, tests that assess arterial inflow or mimic conditions after anastomosis formation and assess the ability of the artery to increase blood flow are also useful, especially in borderline cases.

When severe calcifications are present, significant proximal stenosis should be excluded. Experts recommend that the entire arterial system of the upper limb is screened in a longitudinal view using a color Doppler ultrasound to detect sites of increased velocities and possible stenosis [32]. Not only is this time consuming, but it often cannot be performed in severely calcified arteries as the Doppler signal penetration is often poor and drop outs are present, limiting its use at the very sites where it would be most needed [26]. Therefore, Doppler criteria are needed that can be used at the site of potential anastomosis and can exclude significant proximal stenosis; some of these are discussed below.

#### 4.2.1. Clinical Examination

A good, palpable radial artery pulse is a sign of healthy arterial circulation (a healthy wall as well as good perfusion pressure) and is often considered a prerequisite for AVF construction. Nevertheless, the absence of a radial pulse does not necessarily mean that ipsilateral vascular access is contraindicated [33].

Blood pressure measurements in both arms can detect hemodynamically significant proximal arterial stenosis. A difference in the measured systolic blood pressure of >20 mmHg between both arms indicates a high probability of significant proximal arterial stenosis, which can cause AV fistula non-maturation or hand ischemia and should be treated before AV fistula formation on such an arm [33].

#### 4.2.2. Pulsed-Wave Doppler Assessment

The Doppler curve is usually described by measuring peak systolic velocity (PSV), end diastolic velocity (EDV) and the resulting resistance index (RI = (PSV − EDV)/PSV). The advantage of RI over absolute velocities is that it is less susceptible to measurement error as it is less dependent on the Doppler angle [34]. The acceleration during early systole is described by acceleration time (AT), i.e., the time difference between the onset of systolic upstroke and the top of the first systolic peak [35,36]. To avoid problems with different curve shapes, the maximal systolic acceleration (Acc_max_) can be measured as the steepest part of the systolic upslope, i.e., from the beginning of systole to the point of break in systolic upstroke (i.e., the point at which a change in slope is clearly evident) [35,36]. Some of these parameters, which are routinely used to assess renal artery stenosis, may also be useful in assessing the inflow provided by the feeding artery for future AVF.

##### Peak Systolic Velocity

PSV is associated with the flow at a given arterial diameter. Since the measurements of arterial diameter (and cross-sectional area required for volume flow calculation) are susceptible to measurement error, PSV may be a better alternative to flow measurement. When arterial compliance is reduced due to calcification, PSV is expected to increase, but when significant proximal stenosis develops, PSV decreases distally to the site of stenosis.

Sedlacek et al. used a PSV of >50 cm/s as a cutoff for AVF creation in their study (together with a 2 mm cut-off for the diameter of the artery) and found no difference in PSV in diabetic and nondiabetic patients above this cut-off [37]. Lockhart, who also used only arteries with a diameter of >2 mm, found no difference in the AVF outcomes with a PSV above or below 50 cm/s [38]. Recently, using a much lower diameter cut-off of 1.3 mm and a PSV cut-off of 25 cm/s as criteria for AVF construction, Suresh Kumar reported that the PSV < 45 cm/s was significantly associated with primary AVF failure in a multivariate analysis (OR 8.5 (95% CI 3.0–87.1), *p* = 0.005) [7]. On the other hand, we also did not detect a difference in PSV in patients with successful AVF creation vs. primary failure in a very small group of patients with calcified arteries of an adequate diameter (>2 mm) [27]. Because PSV is influenced by arterial stiffness and calcification, it cannot be considered a reliable factor for fistula prognosis [32,39] and its role in predicting AVF maturation in borderline cases, e.g., small or calcified arteries, remains to be established.

##### Doppler Curve Assessment, Acceleration Parameters

In a pulsed-wave Doppler mode, the shape of the signal obtained should be evaluated. Downstream of a hemodynamically significant stenosis, a poststenotic Doppler pattern is seen with a flattened morphology that loses its typical triphasic shape and becomes monophasic, with parvus (low PSV, low RI) and tardus (slow acceleration (low Acc_max_), long acceleration time (AT)) features [32,40]. PSV is reduced as a result of the flow obstruction caused by stenosis, and EDV is increased as a result of peripheral vasodilatation caused by ischemia, resulting in a reduced RI.

The evaluation of the shape of the Doppler curve to exclude proximal stenosis is a well-established method in other vascular beds. Studies have shown that a poor (blunted, with reduced acceleration in early systole) monophasic duplex waveform in the common femoral artery is an accurate marker of proximal aortoiliac obstructive disease [41] and that Doppler curve analysis (loss of triphasic signal) at the ankle level can be used to rule out peripheral arterial disease in patients prone to mediocalcinosis [42]. A decreased systolic acceleration after significant stenosis can be assessed visually by the shape of the Doppler curve (see Figure 3), but it can also be evaluated objectively by measuring Acc_max_ and AT, as it is in the evaluation of renal artery stenosis [35]. AT was also shown to have diagnostic value in patients with peripheral arterial disease; AT had a high correlation with the toe–brachial index and a high diagnostic accuracy for detecting critical limb ischemia [43]. AT is dependent on the PSV, therefore different cut-off values can be expected in different vascular beds (e.g., a 70 ms cut-off is used for intrarenal Doppler, while 120 and 215 ms cut-offs were suggested for the femoral artery [43,44]). Acc_max_ may be more comparable between different vascular beds, although its measurement requires an optimal insonation angle, while AT measurements are not dependent on the insonation angle [36]. There are no data in the literature on AT in the context of vascular access, but AT, Acc_max_ and AI should definitely be explored in this context in the future, although significant proximal stenosis in the upper extremity is quite rare.

##### Reactive Hyperemia Test

To assess the ability of the artery to provide adequate inflow for AVF, conditions after anastomosis creation can partially be mimicked by a reactive hyperemia test. There are several described protocols for inducing ischemia, ranging from inflating a blood measurement cuff to 50 mmHg above systolic blood pressure for several minutes [45] to simple fist clenching for 2 min [32,38] or even less [27]. The ischemic stimulus induces arterial dilatation through several mechanisms [45]. While the feeding artery itself dilates, which can be measured in larger healthy arteries, a significant portion of the increased flow is due to a decreased peripheral resistance. Therefore, although the reactive hyperemia test likely does not test the ability of the calcified feeding artery to dilate, it might nevertheless assess its ability to provide an increased flow (in response to ischemia or creation of anastomosis). Different Doppler parameters can be measured to evaluate reactive hyperemia, either an increase in PSV or a decrease in RI. Because RI is less susceptible to measurement error than absolute velocities (PSV, EDV) [34], measuring RI during reactive hyperemia may be the best test for a functional assessment of the artery (see Figure 4).

Malovrh showed that a reduced hyperemic response (decrease in RI), defined as a RI during reactive hyperemia ≥0.70, was a predictor of a poor immediate AVF outcome, resulting in a success rate of 40% as compared to 95% in the group with a sufficient response [17]. For AVFs that were patent at 24 h, it was shown that maturation was better at 12 weeks, with a blood flow of about 500 mL/min in patients with a good hyperemic response vs. 300 mL/min in non-responders [17]. Lockhart did not find the RI during reactive hyperemia (nor delta PSV) to be a good predictor of AVF maturation [38]. In his study, the RI during reactive hyperemia was only marginally different (*p* = 0.09) in the matured vs. nonmatured group [38]. Wall found a significant difference in the secondary, but not primary, patency of radial-artery-based AVFs in patients based on their hyperemic response (defined by delta PSV), indicating that accesses based on arteries with acceptable hyperemic responses are more likely to be salvaged by revisions [46], possibly due to poor maturation. There was also no difference in RI during reactive hyperemia in the successful and failed groups in our small group of patients with calcified radial arteries, furthermore, successful AVFs were created in patients with much higher RIs (up to 0.90) than suggested by Malovrh [27]. Therefore, the reactive hyperemia test could be useful in selected patients with borderline arteries, but further prospective studies are needed.

## 5. Outcomes of AVFs Placed on Calcified Arteries

There are relatively few reports in the literature on the outcomes of AVFs placed in patients with significant macroscopic calcifications of the inflow artery and unfortunately, many studies on AVF outcomes do not report the presence of arterial calcifications [47,48]. Even in larger studies, the number of patients in whom significant calcifications are present is small. Mild or moderate calcifications of the radial artery are often not considered a contraindication for AVF creation, while severe calcifications often are [49,50]. 

The available literature on radiocephalic AVF placement on calcified radial arteries is summarized in Table 2. A case-control study by Kim et al. [51] compared 18 patients with mild, spotty calcifications and 29 patients without any calcifications in the inflow artery. Fistulas were mainly brachiocephalic AVFs and patients with a totally circumferential calcification in the inflow artery were excluded from the study. Clinical (93%) and radiological maturation rates (100%) were excellent in the mildly calcified group and comparable to the noncalcified group, but there was a moderate reduction in fistula blood flow [51]. Sedlacek reported a cohort of 25 AVFs placed in patients with an ultrasound finding of vascular calcifications (not graded by severity) and 20 of them (80%) were usable, which was comparable to noncalcified and nondiabetic groups [37]. When AVFs were placed on radial arteries with more severe calcifications, the outcomes were worse. Srivastava reported 15 diabetic patients with calcifications in the radial artery (not graded), of which only 40% matured [13]. Georgiadis reported a cohort of 47 AVFs in patients with linear, railroad track-type macrocalcifications of at least 6 cm of length present on the side of radiocephalic AVF creation [9]. In their cohort, there were two immediate failures and 3 AVFs did not mature [9], from which a maturation rate of 89% can be calculated. The presence of macrocalcifications significantly decreased primary and secondary patency rates, with a reported 1-year secondary patency rate of 52% [9]. We published our results on the radiocephalic AVF outcomes of 18 patients with moderately or severely calcified radial arteries [27]. In 5 patients, anastomosis creation was not possible due to severe calcifications and 1 AVF did not mature, resulting in a maturation rate of 67%, while the 1-year secondary patency rate was 66% [27]. We believe that these acceptable results make the construction of a radiocephalic AVF on a moderately or even severely calcified radial artery a worthwhile attempt if the other conditions (vein and artery diameters) are optimal. 

## 6. Predicting Distal Ischemia in Patients with Calcified Arteries

In addition to the question of whether an AVF can be created and whether it will mature, the complications of AVF placement should also be considered. Hemodialysis access-induced distal ischemia (HAIDI) is a rare but devastating complication, which can develop immediately after vascular access creation or at a later time and can result in loss of tissue or a loss of vascular access. In the Hemodialysis Fistula Maturation Study of a large multicenter prospective cohort of 602 patients, symptoms of HAIDI occurred in 7% of patients and an intervention was required in 4% [52]. Kudlaty reported on a cohort of 303 patients in which a surgical intervention for HAIDI was necessary in 5% of cases, and this occurred only in AVFs/grafts with brachial artery inflow [53]. Morsy reported on a cohort of 299 arteriovenous grafts in which HAIDI occurred in 4.3% of cases, while among 110 forearm AV fistulas, it only occurred in 1.8% of cases [54]. We reported HAIDI requiring AVF ligation in 1% of patients with a functioning AVF after kidney transplantation [55]. HAIDI is therefore rare in distal accesses (about 2%) and more common in brachial artery-based accesses, including grafts (4–5%).

The known patient-related risk factors for the development of HAIDI include the female gender (because women have narrower arteries); risk factors for the presence of atherosclerosis or mediocalcinosis, mainly diabetes mellitus, older age and smoking; and clinically manifested atherosclerosis in other vascular beds, i.e., coronary or peripheral arterial disease [33,56]. Patients with long standing immunosuppression, mainly with corticosteroids, e.g., after a kidney transplantation or patients with systemic vasculitis or resistant glomerular diseases, are also more prone to develop arterial disease and therefore HAIDI. The risk factors for HAIDI from a surgical perspective include proximal anastomosis to the brachial artery and the size of the anastomosis [33,56]. 

There are no simple solutions in patients requiring vascular access, who have arterial calcifications present. Because radiocephalic AVF has the lowest risk for HAIDI, radial artery-based access should be preferred based on the evaluation discussed in this article. If the radial artery at the wrist is too calcified, anastomosis is often performed on a proximal radial artery or brachial artery, which are less calcified due to peripheral distribution of calcifications. An anastomosis to the proximal radial artery has a lower probability of distal ischemia and is the preferred second option [33,57]. Experts suggest that caution should be exercised when using the brachial artery for AVF or graft inflow in patients with calcified arteries, as the probability of HAIDI increases significantly [58,59,60].

Unfortunately, there are no established and reliable methods to predict which patients will develop HAIDI. This issue is particularly important before brachial artery-based access creation in the presence of significant distal radial and ulnar calcifications. In one study of brachial artery based vascular accesses, the development of HAIDI was not related to the access blood flow [61], and in another study, hand ischemia (whether related to vascular access or not) in dialysis patients was almost always the result of arterial disease in the forearm or hand [62]. Therefore, an assessment of the distal arterial circulation in the hand is likely crucial in predicting HAIDI. 

### 6.1. Allen Test

Many centers routinely test for a patent palmar arch with a clinical or Doppler Allen test before radial artery puncturing for interventional procedures, AVF construction or radial artery harvesting for a coronary bypass. However, the actual value and necessity of this test are unclear. Ruengsakulrach examined 50 hands of cadavers and observed that there is always a significant communication between the radial and ulnar arteries in the hand, and therefore concluded that in the absence of vascular disease, harvesting the radial artery should be regarded as a safe procedure [63]. Manabe collected case reports of hand ischemia after radial artery intervention or harvesting and found that an incomplete palmar arch was not documented as a cause of severe hand ischemia, but rather occlusive arterial disease in the forearm arteries [64]. Therefore, Manabe suggests that the preoperative exclusion of occlusive arterial disease in the forearm is essential to avoid severe ischemia of the hand [64]. Radial arteries were harvested for coronary bypasses even in the absence of a normal Allen test, if the Doppler examination of the forearm arteries was normal (i.e., excluding occlusion or significant stenosis, but not proving a patent palmar arch) without ischemic consequences to the hand [65]. Furthermore, no exercise-induced hand ischemia was shown in patients with chronic radial artery occlusion after an interventional procedure [66]. Finally, in some reports of patients with HAIDI requiring surgical revision, all patients had a normal Allen test prior to graft placement, which induced HAIDI [67]. Therefore, a (non)patency of the palmar arch probably does not play a significant role in the development HAIDI.

### 6.2. Finger Pressure and Digital Brachial Index (DBI)

Systolic finger pressure measured by preoperative plethysmography or its ratio to brachial systolic blood pressure, i.e., the digital brachial index (DBI = finger pressure/brachial pressure), was also used to assess distal perfusion and to detect significant arterial disease and therefore the risk of ischemia after vascular access creation. The absolute values of finger pressures or DBI were not found to be very useful. There was only a borderline difference in DBI before a brachial artery-based AVF/graft creation in patients who developed HAIDI and those who did not (0.93 ± 0.18 vs. 1.0 ± 0.14, *p* = 0.08), resulting in relatively low sensitivity (64%) and specificity (69%) [68]. The authors concluded that patients with a preoperative DBI <1.0 are more likely to develop steal syndrome, but there was no DBI threshold below which steal is inevitable [68]. In another study, there was no difference in the preoperative DBI in patients with or without HAIDI after brachial artery-based access [61]. Therefore, the American Society of Diagnostic and Interventional Nephrology, in its white paper on the assessment and management of hemodialysis access-induced distal ischemia, concluded that although DBI is useful for risk assessment, there is no value at which the development of HAIDI is inevitable [69]. 

On the other hand, a recent study of 105 patients before primary access creation also found that patients who developed severe HAIDI had comparable systolic finger pressure and DBI preoperatively (with a trend to higher values), but there was an almost two-fold greater drop in finger pressure after radial artery compression [70]. A 40 mmHg drop in finger pressure was predictive of a ten-fold higher risk of developing severe HAIDI; 6 of 10 HAIDI cases occurred after brachiocephalic AVF construction [70]. Therefore, the change in finger pressure with radial artery compression may be more predictive of distal ischemia than the absolute values of finger pressure or DBI. The disadvantage of plethysmography is that it is time-consuming and cannot be performed when the finger arteries are incompressible due to severe mediocalcinosis. Interestingly, the incompressibility of the finger arteries did not increase the incidence of HAIDI in a small cohort of patients [70].

### 6.3. Ultrasound and Doppler Examination

When considering the placement of an AVF on a calcified radial artery, the presence and patency of the ulnar artery as well as the adequacy of its Doppler signal should be confirmed. The presence of calcifications in the ulnar artery, which are always present in the case of radial artery calcifications, should not be considered a contraindication by itself. 

As discussed above, the analysis of the Doppler curve shape of the radial artery, that is, the assessment of maximal acceleration during early systole (Acc_max_), can likely detect hemodynamically significant proximal arterial stenosis, which should be treated before, or very soon after ipsilateral vascular access creation at any level. How best to predict distal ischemia with brachial artery-based access remains to be established, but an adequate PSV in the radial artery could be the equivalent of digital pressure.

## 7. Conclusions

The methods of detecting arterial calcifications, assessing the suitability of calcified arteries for providing inflow before the construction of an AVF and possible predictors of distal ischemia described in this review, along with their advantages and disadvantages in clinical practice, are summarized in Table 3, Table 4 and Table 5. A noninvasive ultrasound examination is probably the best tool for a morphologic and functional assessment of the arteries, as it is more objective, provides several parameters that can be assessed and enables a detailed assessment of the anastomotic site itself. Overall, the outcomes of forearm AVFs with calcified arteries are not poor, and in our opinion, a fifty-fifty maturation probability still makes an AVF attempt worthwhile.

### 7.1. Predicting the Difficulty of Clamping and Surgical Procedure

In the third millennium, plain radiographs and clinical assessments should be left for the low-income countries, where they can be a valuable tool. The ultrasound assessment of the severity of calcifications should be the standard. A radial artery should probably be dismissed as too calcified when there are severe calcifications present (see Table 1) with a very patchy color Doppler and the artery is incompressible with the ultrasound probe. These may be the best predictors that the artery can be clamped and the anastomosis sutured without extreme surgical measures, although further studies are needed. 

### 7.2. Predicting Successful Maturation

Blood pressure measurements in both arms and good acceleration during systole in the Doppler exam can probably exclude hemodynamically significant proximal arterial stenosis, which does not represent an absolute contraindication for AVF placement but should be evaluated and treated either before or after AVF placement. It is conceivable that any radial artery of suitable diameter (>2 mm), regardless of the severity of calcifications, can provide sufficient inflow for an AVF if the anastomosis can be made and an above-average vein is present. If this is not the case, the RI during reactive hyperemia seems to be the best simulation of the conditions after anastomosis formation, and therefore, for the functional assessment of the artery, although the cut-off value remains to be established and we have not yet seen an artery without a hyperemic response. Therefore, the best parameter for this assessment before surgery remains to be established.

### 7.3. Predicting Distal Ischemia

If a suitable vein is present, it is always best to make the anastomosis to the radial artery, at least in the proximal part. If a suitable forearm vein is not available and the anastomosis must be made at the level of the elbow, the question remains of how to predict distal ischemia in cases of severe calcifications in the distal arteries. There are no good predictors of a very high probability of distal ischemia, so this remains an unresolved problem that requires further study. Most likely, it is not reasonable to attempt elbow AVF or graft when a single calcified artery is present in the forearm or if the PSV is low.

## Figures and Tables

**Figure 1 diagnostics-13-02660-f001:**
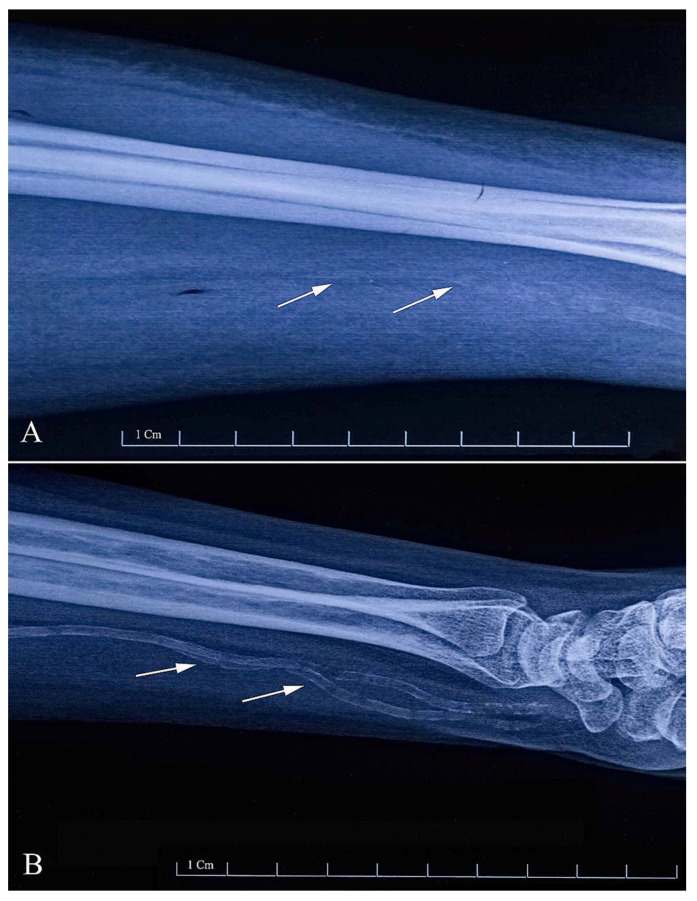
Plain forearm radiographs with visible arterial calcifications (arrows, graded as <6 cm in length, panel (**A**), and >6 cm, panel (**B**); reproduced from [8] (CC BY 4.0)).

**Figure 2 diagnostics-13-02660-f002:**
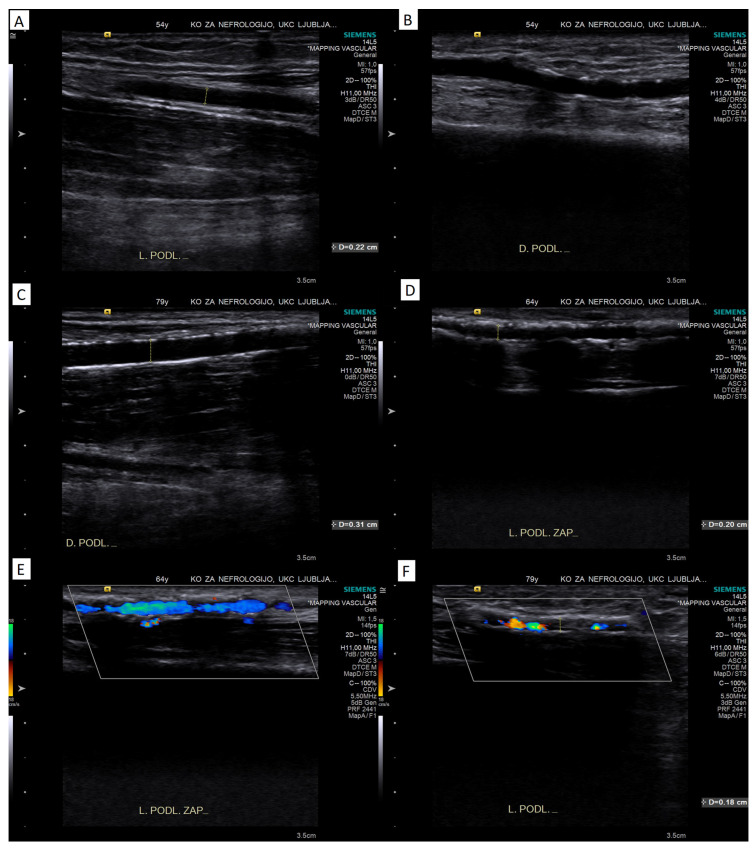
Illustrative B-mode and color Doppler images of the proposed grading of severity of calcifications. Panel (**A**): no calcifications (clear separation of intima and media); panel (**B**): mild calcifications (increased echogenicity/spotty calcifications without distal shadowing); panel (**C**): moderate calcifications (linear calcifications with incomplete distal shadowing); panel (**D**): severe calcifications (diminished separation from surrounding tissue, continuous calcification with distal shadowing); panel (**E**): partly homogenous color Doppler signal in a calcified artery; panel (**F**): very patchy color Doppler signal in a calcified artery.

**Figure 3 diagnostics-13-02660-f003:**
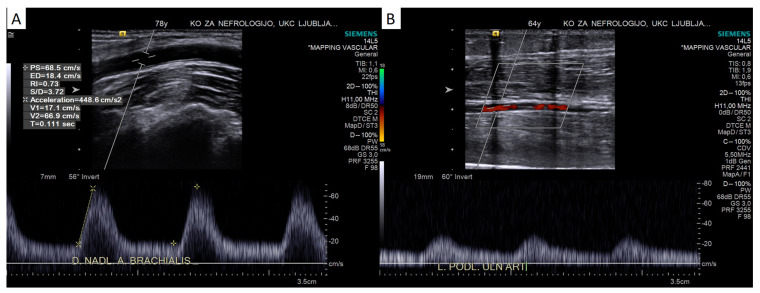
Monophasic pulsed-wave Doppler waveform in the brachial artery with prolonged acceleration time (AT 111 ms, Acc_max_ 4.5 m/s^2^, panel (**A**)) in patient with significant stenosis of subclavian artery. An even more dampened signal in the radial artery (panel (**B**)) of another patient with occluded subclavian artery. For comparison, a normal triphasic signal is presented in Figure 4A.

**Figure 4 diagnostics-13-02660-f004:**
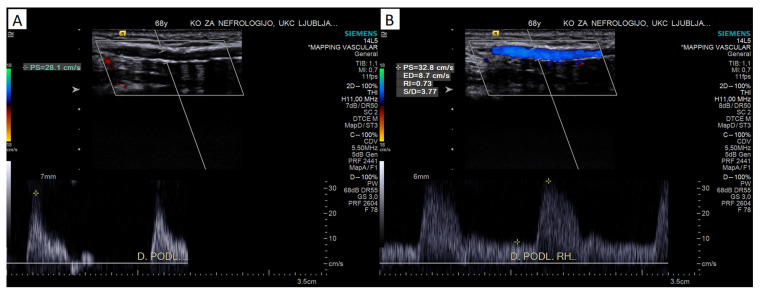
Reactive hyperemia test with an excellent response (increase in diastolic flow/velocity) in a moderately calcified radial artery: reduction in resistance index from 1.0 at rest (panel (**A**)) to 0.73 at reactive hyperemia (panel (**B**)).

**Table 1 diagnostics-13-02660-t001:** Proposed grading of severity of arterial calcifications based on B-mode and color Doppler ultrasound findings, adapted from [1,17,26,27].

Calcification Grade	B-Mode Image	Color Doppler Image	Appropriateness for Fistula Creation
none	smooth vessel wall, clear separation of intima and media	homogenous signal	yes
mild	minor wall structure irregularities, increased echogenicity with spotty calcifications but without distal shadowing	homogenous signal	yes
moderate	irregular wall structure, intermittent calcifications with distal shadowing or linear calcifications with incomplete distal shadowing	partly homogenous signal (drop outs <50% of visible artery length)	likely
severe	irregular wall structure with diminished separation from surrounding tissue, continuous calcification of the wall with distal shadowing	very patchy (drop outs >50% of visible artery length) or almost absent signal	Not likely/careful consideration

**Table 2 diagnostics-13-02660-t002:** Outcomes of radiocephalic AVFs placed on calcified radial arteries.

Reference	N in the Calcified Group	Degree of Calcifications	Clinical Maturation Rate	1-Year Secondary Patency Rate
Sedlacek, 2001 [37]	25	not graded	80%	/
Georgiadis, 2014 [9]	47	moderate?	89% *	52%
Srivastava, 2018 [13]	17	not graded	48%	/
Kim, 2019 [51]	18	mild (spoty)	93%	/
Suresh Kumar, 2019 [7]	9	moderate/severe?	22%	/
Sadasivan, 2021 [8]	11 3	mild/moderate severe	73% 33%	/ /
Gubensek, 2022 [27]	18	moderate/severe	67%	66%

* Calculated as 42/47 (2 immediate failures, 3 non-matured AVFs) [9].

**Table 3 diagnostics-13-02660-t003:** Overview of methods for assessment of calcifications of the inflow artery, their advantages and disadvantages.

Method	Advantages	Disadvantages
clinical examination (palpation of the arterial wall, presence of the pulse)	simple, no equipment needed, predictive of primary AVF failure [7]	subjective, no grading of severity of calcifications, no assessment of the site of anastomosis
plain radiograph	simple, predictive of primary AVF failure [7,9]	patient irradiation, no grading of severity of calcifications at the site of anastomosis
B-mode and color Doppler ultrasounds (presence and grading of calcifications, compressibility of the artery)	calcifications severity grading possible, assessment of the site of anastomosis, predictive of functional maturation [13]	partially subjective, depends on operator and experience

AVF—arterio-venous fistula.

**Table 4 diagnostics-13-02660-t004:** Overview of methods for functional assessment of the ability of the artery to provide sufficient inflow and to exclude significant proximal stenosis. Their advantages and disadvantages are given.

Method	Advantages	Disadvantages
clinical examination (presence of the pulse, difference in blood pressure on both arms)	simple	moderate sensitivity for detection of proximal stenosis
Doppler ultrasound (PSV, assessment of the shape of doppler curve, AT, Acc_max_)	objective, successfully used in other vascular beds to exclude significant proximal stenosis	moderately time consuming, no cut-off values validated in the field of vascular access
Doppler ultrasound (RI during reactive hyperemia)	functional test, mimics the conditions after AVF creation	time consuming, no clear cut-off value, effect on AVF outcome unclear [17,27,38]

Acc_max_—maximal systolic acceleration, AT—acceleration time, AVF—arterio-venous fistula, PSV—peak systolic velocity.

**Table 5 diagnostics-13-02660-t005:** Overview of methods for predicting distal ischemia after arterio-venous fistula construction in the arm with calcified arteries.

Method	Advantages	Disadvantages
clinical examination (Allen test, presence of the pulse)	simple	likely not predictive of distal ischemia [67]
plethysmography (finger pressure, DBI, delta finger pressure)	objective, used in other vascular beds	time consuming, no clear cut-off value, predictive value uncertain [61,68,70]
Doppler ultrasound	objective, likely an alternative to plethysmography	no parameters established yet

DBI—digital brachial index, PSV—peak systolic velocity.

## Data Availability

Data sharing not applicable.

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
