# Peer review of "The Role of Ultrasound Examination in the Assessment of Suitability of Calcified Arteries for Vascular Access Creation—Mini Review"

_diagnostics, 2023, doi:10.3390/diagnostics13162660_

Round 1

Reviewer 1 Report

In the review, “ The Role of Ultrasound Examination in the Assessment of Suitability of Calcified Arteries for Vascular Access Creation”, the authors discussed the potential association of arterial calcifications with arterio-venous fistula (AVF) formation and maturation, as well as the related complications, such as distal ischemia. This review listed the methods for detecting arterial calcifications and discussed the suitability of calcified arteries for AVF creation and its possible outcome. The methods including radiography, ultrasound, and functional tests for the artery function have been summarized. The author also discussed the development of distal ischemia after creation of AVFs. They concluded that noninvasive ultrasound examination is probably the best tool for morphologic and functional assessment of the arteries. This review was well-written and organized, however, the following two questions should be discussed in the Review.

1. Is AVF calcification linked with artery calcification?

2. Does AVF calcification deteriorate AVF maturation?

Author Response

Reviewer 1

"In the review, “ The Role of Ultrasound Examination in the Assessment of Suitability of Calcified Arteries for Vascular Access Creation”, the authors discussed the potential association of arterial calcifications with arterio-venous fistula (AVF) formation and maturation, as well as the related complications, such as distal ischemia. This review listed the methods for detecting arterial calcifications and discussed the suitability of calcified arteries for AVF creation and its possible outcome. The methods including radiography, ultrasound, and functional tests for the artery function have been summarized. The author also discussed the development of distal ischemia after creation of AVFs. They concluded that noninvasive ultrasound examination is probably the best tool for morphologic and functional assessment of the arteries. This review was well-written and organized, however, the following two questions should be discussed in the Review.

  1. Is AVF calcification linked with artery calcification?

This review does not discuss general (systemic) vascular calcifications in CKD patients, but focuses on the calcifications of the feeding arteries of the arm and their influence on AVF outcomes (i.e. how can a significantly calcified artery still provide sufficient inflow to an AVF). Both are of course linked as calcifications are a systemic condition, although with peripheral predominance, but such pathophysiological discussion is beyond the scope of this review. The calcifications, sometimes present in an older AVF (i.e. the fistula vein) itself as part of degeneration of the vessel wall are also not discussed, as these are not clinically very important.

  1. Does AVF calcification deteriorate AVF maturation? "

Feeding artery calcifications do adversely affect AVF maturation and this is already discussed in the manuscript (under subtitle "5. Outcomes of AVFs placed on calcified arteries") and the effect of various parameters of arterial assessment on AVF outcomes are also discussed with each separate parameter, when the literature on this exists. How general, systemic calcifications (aorta, coronary, etc.) affect AVF outcome is not discussed, as we can assess the feeding artery itself, which should provide inflow to the AVF, and this has a much greater predictive value than a general assessment of arterial calcifications.

Reviewer 2 Report

This mini review describes the methods detecting arterial calcifications to assess the suitability of calcified arteries before constructing an AVF. The review is well written. However, this reviewer has a few comments.

o   The significance of writing this review and how this may be helpful for the clinical staff and researchers should be clear in the introduction. Also, given that all the methods to some extents are already in the literature, it should be clear what new message/direction this review brings to the readers.

o   It would be good to include a section writing imaging technique to assess vascular calcification.

 o   In addition, for a comparative study, all the methods can be mentioned in a table with their uses, advantages, and disadvantages.

Author Response

"This mini review describes the methods detecting arterial calcifications to assess the suitability of calcified arteries before constructing an AVF. The review is well written. However, this reviewer has a few comments.

o   The significance of writing this review and how this may be helpful for the clinical staff and researchers should be clear in the introduction. Also, given that all the methods to some extents are already in the literature, it should be clear what new message/direction this review brings to the readers.

Thank you for this insightful comment. I have added two paragraphs (one moved from the conclusions) to the introduction:

"Precise evaluation of the suitability of a calcified artery for possible creation of an AVF remains a challenging task. This task will become increasingly more important in the future as the ESKD population is getting older and calcifications become even more prevalent. Furthermore, the decision to perform a forearm AVF instead of an elbow AVF whenever possible, even in the presence of calcifications, is not only an academic or policy issue, because forearm AVFs have much lower frequency of ischemic complications. Therefore, arterial assessment is of utmost importance if we are to create distal radio-cephalic AVFs in our elderly patients whenever possible without burdening them with futile surgical attempts.

This review is intended for nephrologists, working in the field of vascular access and interventional nephrology, and focuses on methods to detect arterial calcifications and to assess the suitability of calcified arteries for providing inflow before construction of an AVF. It gives a critical appraisal of existing methods and some comparison between the fields of vascular access and peripheral arterial disease. Ultrasound examination is the main tool we have and a grading system is proposed for quantifying the severity of calcifications in the arteries of the arm with B-mode and Doppler ultrasound exam. Some of Doppler parameters described in the literature, which can be used for arterial as-sessment, are discussed and the best parameters that can be used in clinical practice are emphasized. Possible predictors of the development of distal ischemia are also discussed, as well as overall outcomes of AVFs placed on calcified arteries."

o   It would be good to include a section writing imaging technique to assess vascular calcification.

I do not fully understand the suggestion of the reviewer. This review does not discuss general (systemic) vascular calcifications in CKD patients and their imaging/detection techniques (abdominal X-ray, coronary CT, etc.), but focuses on the calcifications of the feeding arteries of the arm and their influence on AVF outcomes. Imaging techniques (plain radiograph, B-mode and also color doppler assessment) are discussed in the manuscript and a grading system, based on the literature, is proposed (Table 1).

 o   In addition, for a comparative study, all the methods can be mentioned in a table with their uses, advantages, and disadvantages. "

Thank you for this nice suggestion. The methods described for the three different purposes (grading severity of calcifications, functional assessment and distal ischemia prediction) and their (dis)advantages are now summed up in Tables 3-5, which were added at conclusion of the article.